# Evolving Precision First-Line Systemic Treatment for Patients with Unresectable Non-Small Cell Lung Cancer

**DOI:** 10.3390/cancers16132350

**Published:** 2024-06-26

**Authors:** Tianhong Li, Weijie Ma, Ebaa Al-Obeidi

**Affiliations:** 1Division of Hematology/Oncology, Department of Internal Medicine, University of California Davis School of Medicine, University of California Davis Comprehensive Cancer Center, Sacramento, CA 95817, USA; weijie.ma@hitchcock.org (W.M.);; 2Medical Service, Hematology/Oncology, Veterans Affairs Northern California Health Care System, 10535 Hospital Way, Mather, CA 95655, USA; 3Department of Pathology and Laboratory Medicine, Dartmouth Hitchcock Medical Center, Geisel School of Medicine at Dartmouth, Lebanon, NH 03756, USA

**Keywords:** NSCLC, first-line therapy, biomarkers, molecular targets, immune biomarkers, immune checkpoint inhibitors, PD-1, PD-L1, CTLA-4

## Abstract

**Simple Summary:**

The armamentarium of first-line systemic therapy for patients with unresectable non-small cell lung cancer (NSCLC) has been rapidly expanding over the past two decades. Currently, about 50% of patients with NSCLC do not need chemotherapy as first-line treatment. For these patients, molecularly targeted therapy and immune checkpoint inhibitor therapy have significantly improved progression-free survival and overall survival with favorable toxicity profiles. Patient demographics and clinical features are not sufficient for the selection of these treatments. An unmet need of precision oncology is to select the most appropriate therapeutic regimen with maximal efficacy and minimal unwanted toxicity for individual patients with NSCLC based on the assessment of their molecular and immune biomarkers at diagnosis. In this review, we summarize the current data and propose a practical algorithm for implementing precision biomarker testing at diagnosis and selecting the most appropriate first-line systemic therapy for patients with unresectable, advanced, or metastatic NSCLC.

**Abstract:**

First-line systemic therapy for patients with advanced or metastatic non-small cell lung cancer (NSCLC) has rapidly evolved over the past two decades. First, molecularly targeted therapy for a growing number of *gain-of-function* molecular targets has been shown to improve progression-free survival (PFS) and overall survival (OS) with favorable toxicity profiles compared to platinum-containing chemotherapy and can be given as first-line systemic therapy in ~25% of patients with NSCLC. Actionable genetic alterations include EGFR, BRAF V600E, and MET exon 14 splicing site-sensitizing mutations, as well as ALK-, ROS1-, RET-, and NTRK-gene fusions. Secondly, inhibitors of programmed cell death protein 1 or its ligand 1 (PD-1/L1) such as pembrolizumab, atezolizumab, or cemiplimab monotherapy have become a standard of care for ~25% of patients with NSCLC whose tumors have high PD-L1 expression (total proportion score (TPS) ≥50%) and no sensitizing EGFR/ALK alterations. Lastly, for the remaining ~50% of patients who are fit and whose tumors have no or low PD-L1 expression (TPS of 0–49%) and no sensitizing EGFR/ALK aberrations, platinum-containing chemotherapy with the addition of a PD-1/L1 inhibitor alone or in combination of a cytotoxic T-lymphocyte-associated protein 4 (CTLA-4) inhibitor improves PFS and OS compared to chemotherapy alone. The objectives of this review are to summarize the current data and perspectives on first-line systemic treatment in patients with unresectable NSCLC and propose a practical algorithm for implementing precision biomarker testing at diagnosis.

## 1. Introduction

Lung cancer contributes to a significant global disease burden with an estimated 2.5 million new cases and 1.8 million deaths in 2022 [1]. In the United States, lung cancer remains the leading cause of cancer death in both men and women despite the fact that lung cancer incidence has steadily declined since 2006 by 2.5% annually in men and by 1% annually in women [2]. The survival rate of lung cancer differs by histologic subtype, stage at diagnosis, access to care, and treatment, which have significant geographic variation. Non-small cell lung cancer (NSCLC) is the most common histology, which comprises 80–85% of lung cancer. Patients with metastatic lung cancer contribute to the majority of lung cancer incidence and death. Despite the benefits of smoking cessation, pollution mitigation efforts, and early detection, over 40% of patients with NSCLC are diagnosed with distant metastasis [2]. Palliative systemic therapy is the main treatment option for both metastatic and locally advanced disease. The 5-year survival is 36% and 0–26% for patients with NSCLC with stage IIIA and IIIB-IV, respectively. In 1995, platinum-based doublet chemotherapy became the mainstay of first-line systemic therapy with improved overall survival (OS) over best supportive care [3]. Since the approval of molecularly targeted therapy, the incidence-based mortality from NSCLC decreased by 6.3% annually from 2013 through 2016 per the U.S. Surveillance, Epidemiology, and End Results (SEER) 18-registry database [4]. Between 1990 and 2014, 1-year relative survival of Californian patients with metastatic lung cancer was significantly improved from 18.4% to 29.4%, and 5-year relative survival significantly improved from 2.2% to 5.0% [5]. This survival improvement was mostly observed in patients with metastatic lung adenocarcinoma in California due to the uptake of tumor genotyping and first- and second-generation molecularly targeted therapies. Furthermore, with the United States Food and Drug Administration (U.S. FDA) approval of immune checkpoint inhibitor (ICI) therapy targeting programmed cell-death protein 1 (PD-1) or its ligand 1 (PD-L1) in 2015, there has been further improvement in lung cancer-specific survival [6,7]. Forty-three cancer drugs have been approved by the U.S. FDA since the 2000s, with 35 being approved since 2015 [8]. Figure 1 summarizes the major milestones for patients with advanced or metastatic NSCLC in the past few decades. This armamentarium of lung cancer therapeutics has enabled the current era of precision lung cancer care, leading to improved progression-free survival (PFS), OS, and quality of life. In this review, we summarize the key data and propose an integrative algorithm for the workup and selection of first-line systemic treatment for patients with nonresectable, locally advanced or metastatic NSCLC.

## 2. Histopathological Diagnosis

NSCLC has a broad spectrum of molecular, genomic, and immunological heterogeneity [7]. The intricate interplay of molecular aberrations orchestrates tumorigenesis and the dynamic phases of immunoediting, characterized by the elimination, equilibrium, and escape of neoplastic cells from immune surveillance, culminating in metastatic dissemination [9]. The cornerstone of NSCLC management, especially in the metastatic setting, lies in the precision of its tissue diagnosis. The usual diagnostic panel typically includes a combination of histopathological examination, immunohistochemistry (IHC), and molecular and immune biomarker testing. NSCLC is categorized into several major histological subtypes, including adenocarcinoma (LUAD), squamous cell carcinoma (LUSC), large cell neuroendocrine tumor (LCNET), and other less common subtypes including adenosquamous carcinoma, sarcomatoid carcinoma, etc. Each histology subtype has distinctive pathological features and therapeutic implications, especially in selecting systemic chemotherapy. For patients with advanced lung cancer, the majority (70–90%) of histopathological diagnosis is made on tumor specimens from a small biopsy [10]. LUAD is subclassified into several histological subtypes (such as lepidic, papillary, and acinar subtypes) according to the latest World Health Organization (WHO) classification of lung carcinoma from 2015 [11]. A strategic IHC panel comprising markers for LUAD (e.g., thyroid transcription factor-1 [TTF-1] and napsin A) and LUSC (e.g., p40 and p63) is sufficient for histology subtype identification [12]. The nuclear expression of TTF-1 is present in 70–90% of LUAD and absent in LUSC. The granular cytoplasmic expression of napsin A is observed in 80–85% of LUAD. The combination of TTF-1 and napsin A IHC expression has a sensitivity of 70–90% and a specificity of 90–100% for the diagnosis of LUAD [13,14]. In contrast, the specificity of p40 and p63 expression is 95–98% and 60–75%, respectively, for the diagnosis of LUSC [12,15]. The sensitivity of p40 and p63 expression is 77–80% and 86–90%, respectively [16,17]. For rare subtypes, diagnosis is established through comprehensive morphological evaluations supplemented by ancillary and molecular studies. For example, in the case of sarcomatoid carcinomas, mesenchymal markers such as S100, myogenin, and MYOD1 are used for staining, which varies according to the specific subtype. In 2021, a new WHO classification of thoracic tumors of pulmonary epithelial origin is defined as SMARCA4-deficient undifferentiated tumor (SMARCA4-UT), which has distinct histopathological features and loss-of-function genomic alterations of the SMARCA4 gene [18]. The histologic diagnosis of LUAD, LUSC, and other subtypes is essential for selecting systemic chemotherapy for NSCLC. Currently, there is no specific treatment recommendation for subtypes of adenocarcinomas or SMARCA4-UT. 

## 3. Molecular Biomarker Testing

Cancer is a disease of accumulative aberrant genome alterations. Patients with LA/mNSCLC with driver oncogenes often present with a large tumor burden and symptoms that warrant prompt systemic therapy. It is important to quickly identify those patients with NSCLC with actionable driver oncogenes for molecularly targeted therapy at initial diagnosis. Companion diagnostics in NSCLC use different testing techniques, including real-time polymerase chain reaction (RT-PCR), fluorescence in situ hybridization (FISH), IHC, and next-generation sequencing (NGS) of multiplex or targeted gene panels [19]. NGS can simultaneously detect single nucleotide variants (SNVs), copy number variations (CNVs), and structural rearrangements such as gene fusions, leading to a high diagnostic yield of actionable genomic alterations in tumor specimens. Increasingly, these molecular biomarker assays have been used for selecting initial and subsequent therapy based on the tumor genomic makeups and clonal evolution of drug-sensitive or resistant mutations for either molecularly targeted therapy or ICI therapy [20,21]. The College of American Pathologists (CAP), the International Association for the Study of Lung Cancer (IASLC), and the Association for Molecular Pathology (AMP) first issued a molecular testing guideline in 2013 to recommend testing all patients with advanced lung cancer for TKI-sensitive epidermal growth factor receptor (EGFR) and anaplastic lymphoma kinase (ALK) genomic alterations regardless of clinical characteristics, such as age, race, and smoking status [22]. The molecular targets were expanded to include ROS1 and BRAF by different societies in 2018 [23,24,25]. RET, MET, ERBB2 (HER2), and KRAS testing is also recommended if the previous four genes are negative. The tiered testing approach which takes into account the decreasing prevalence of the biomarkers is a reasonable and cost-effective approach that allows clinicians to defer further testing if a more prevalent oncogene mutation, i.e., sensitizing EGFR Exon 19 deletion or L858R mutant, is present. However, the tiered approach requires more tumor specimens due to the sequential testing and likely takes longer compared to NGS. Broad molecular profiling is defined as molecular testing that identifies all biomarkers identified in NSCLC in either a single assay or a combination of a limited number of assays and optimally also identifies emerging biomarkers. In the 2019 National Comprehensive Cancer Network (NCCN) version 2 update, molecular testing for eight actionable molecular alterations (including somatic mutations or rearrangements) was recommended: EGFR, ALK, ROS1, BRAF, MET, RET, ERBB2, and NTRK1/2/3 genes, in patients with non-squamous NSCLC [26]. Since then, NCCN guidelines endorse broad molecular profiling with the goal of identifying rare driver mutations that include high-level MET amplification or MET exon 14 skipping mutations, RET rearrangements, ERBB2 (HER2) mutations, and KRAS G12C mutations for which effective drugs are readily available or to appropriately counsel patients regarding the availability of clinical trials. For those patients in whom broad panel testing has not identified any driver oncogenes, RNA-based NGS can be performed to maximize the detection of fusion events. If molecular testing cannot be performed due to insufficient tissue (i.e., less than 20% tumor cellularity), re-biopsy and re-testing are likely to cause further delays in initiating treatment. Alternatively, liquid biopsy and genomic profiling assays of plasma circulating tumor DNA (ctDNA) have been recommended by the CAP/IASLC/AMP guidelines and are increasingly used in the clinic when tumor tissue is limited and/or insufficient for molecular testing [23,27,28,29]. As the blood contains circulating plasma ctDNA shed by both primary and metastatic tumor sites, tumor genomic profiling of plasma ctDNA might reveal a broader profile of tumor genomic alterations with a shorter turn-around time compared to that of tumor tissue in individual patients. Thus, plasma tumor mutation profiling might provide complementary information to tissue tumor mutation profiling for guiding therapeutic decisions in patients with advanced or metastatic NSCLC, especially those patients with multiple metastases. A meta-analysis performed on more than 30,000 patients suggests that matched personalized therapy improves the outcome across cancer types and studies [30].

## 4. Immune Biomarkers

Cancer is also a disease of immune evasion. The intricate interplay between tumor cells and the immune system reveals that tumor cells can evade destruction by the immune system through a range of complex and often overlapping mechanisms. Among the immune escape mechanisms, the upregulation of immune checkpoints like cytotoxic T-lymphocyte antigen 4 (CTLA-4) in lymphoid organs and PD-1/PD-L1 in peripheral tumor tissues is the most well-studied tumor escape mechanism in many tumor types, including NSCLC. PD-L1 expression, assessed by IHC stain 22C3 (Dako) antibody, received U.S. FDA approval for selecting pembrolizumab as first-line and second-line monotherapy in patients with advanced NSCLC [31]. In addition to PD-L1, several tumor molecular characteristics have been associated with an increased clinical response to PD-1/PD-L1 inhibitors in NSCLC. On June 16, 2020, the FDA granted accelerated approval to pembrolizumab for the treatment of adult and pediatric patients with unresectable or metastatic solid tumors with a high tumor mutational burden (TMB-H) (≥10 mutations/megabase (mut/Mb)), as determined by an FDA-approved test. This approval is for patients who have progressed following prior treatment and have no satisfactory alternative treatment options [32]. These immune biomarkers predict less than 50% of clinical responses, which is lower compared to molecular biomarkers such as activating EGFR or ALK genomic alterations that predict 60–80% of clinical responses to small molecular tyrosine kinase inhibitors. In addition, NGS can also detect microsatellite instability (MSI) arising from defective mismatch repair (dMMR) in patients with metastatic solid tumors, which predicts clinical response to ICIs. However, the prevalence of these genomic alterations is rare in NSCLC [33]. Of note, the utility of existing immune biomarkers varies significantly among different ICIs, tumor types, and the status of companion versus complementary diagnostics. It is essential to develop and validate immune-oncology biomarker assays to select appropriate patients for cancer immunotherapy and monitor treatment response and immune-related adverse effects (irAEs). Recent studies have shown that, in addition to T cells, other immune cell types and blood-based biomarkers have the potential to predict clinical responses to ICIs [34,35,36]. High-throughput multiplexed immunohistochemistry (mIHC) is a promising clinical assay for the simultaneous detection of multiple tumor and immune biomarkers in the tumor microenvironment on a single tissue section [37]. Novel strategies are needed to integrate the real-time assessment of both molecular and immune biomarkers using tissue and blood specimens simultaneously and/or longitudinally. This will help select precision cancer care tailored to individual patients throughout their disease course. 

## 5. Selection of First-Line Systemic Therapy for Patients with Advanced NSCLC

The selection of systemic therapies is based on the tumor’s histology, molecular profiling, and immune biomarkers [26]. Other factors influencing therapy choices for an individual patient include age, comorbidities, organ function, performance status, and treatment preferences. 

### 5.1. First-Line Systemic Therapy for Oncogene-Driven NSCLC

Small molecular TKIs targeting *gain-of-function* molecular targets have been shown to improve PFS and OS with favorable toxicity profiles compared to platinum-containing combination chemotherapy as first-line systemic therapy. As summarized in Figure 2, approximately 25% of patients with NSCLC have actionable genetic alterations, including EGFR-, BRAF V600E-, MET exon 14 splicing site-, ERBB2 (HER2)-, KRAS G12C-sensitizing mutations, and ALK-, ROS1-, RET-, and NTRK-gene fusions. Although PD-L1 expression can be increased in these patients, ICIs should be avoided in these patients with NSCLC with driver oncogenes due to lack of efficacy and increased pulmonary and/or hepatic toxicity, either as single agents or in combination with TKIs [38,39,40]. Therefore, current NCCN guidelines recommend against the use of ICIs in these patients with oncogene-driven NSCLC outside of clinical trials. 

EGFR mutations were the first and among the most prevalent molecular targets identified in patients with LA/mNSCLC. These mutations occur in 30–35% of East Asian and 10–15% of White patients. EGFR Exon 19 deletion (40–50% of cases) and Exon 21 L858R substitution (30–40% of cases) are the two major subtypes of EGFR mutations. The remaining 14% of mutations are termed uncommon EGFR mutations and consist of exon 20 insertions (5%) and other mutations in exons 18 to 21, with representative examples being exon 18 G719X, exon 20 S768I, and exon 21 L861Q. Approximately 14% of uncommon EGFR mutations coexist with other EGFR mutations, either with other common EGFR mutations or with other uncommon EGFR mutations, and together they are termed compound mutations. Erlotinib, a first-generation EGFR TKI, was the first molecularly targeted therapy to receive FDA approval in 2004 as a second-line unselected therapy [41]. The second-generation EGFR TKI afatinib was approved in 2013 [42], while necitumumab in combination with chemotherapy received approval for patients with squamous cell carcinoma in 2015 [43]. The third-generation EGFR TKI osimertinib has shown improved PFS compared to erlotinib and gefitinib [44]. Common EGFR mutations exhibit sensitivity to EGFR TKIs, with an objective response rate (ORR) of 60–80% and a PFS ranging from 9 to 19 months. Due to its favorable toxicity profile, osimertinib has been endorsed as the preferred frontline agent by the NCCN guideline (Version 1, 2020) [45]. Afatinib received FDA approval for three additional EGFR mutations (L861Q, G719X, and S768I) in January 2018, making it the EGFR TKI with the broadest first-line indication in EGFR mutant NSCLC [46]. Recent data have shown that osimertinib has an ORR of 55%, a median PFS of 9.4 months, and a median duration of response (DoR) of 22.7 months in rare, uncommon EGFR mutations other than exon 20 insertions (UNICORN) [47,48].

Patients with EGFR exon 20 insertion mutations are resistant to osimertinib. Several medications have been developed for these patients. On 21 May 2021, the FDA granted accelerated approval to amivantamab-vmjw for adult patients with locally advanced or metastatic NSCLC with EGFR exon 20 insertion mutations that progressed on or after platinum-based chemotherapy [49]. Oral EGFR TKI mobocertinib was initially approved on 15 September 2021 for adult patients with NSCLC with EGFR exon 20 insertion mutations whose disease had progressed on or after platinum-based chemotherapy. However, mobocertinib failed to show significant improvement in median PFS in phase 3 trials and was withdrawn by the manufacturer from use in the US in October 2023. On 1 March 2024, the FDA approved amivantamab-vmjw with carboplatin and pemetrexed for the first-line treatment of LA/mNSCLC with EGFR exon 20 insertion mutations [50]. 

Bevacizumab in combination with carboplatin and paclitaxel chemotherapy received FDA approval as the first-line treatment for patients with unresectable, locally advanced, recurrent or metastatic non-squamous NSCLC in October 2006 [51]. Platinum-based chemotherapy without or with bevacizumab is currently recommended for first-line treatment for HER2 genomic alterations or KRAS G12C mutations. First-generation EGFR TKIs failed to show survival benefit in patients with EGFR mutant advanced NSCLC in several phase III studies [52,53,54,55]. Osimertinib with platinum-based chemotherapy received accelerated FDA approval for patients with LA/mNSCLC with EGFR exon 19 deletions or exon 21 L858R mutations on 16 February 2024. Despite these advances, patients with EGFR-mutant metastatic NSCLC inevitably have tumor recurrence and ultimately succumb to metastatic disease. Many strategies are being explored to overcome the resistance to EGFR inhibitors in NSCLC. 

The second most common genomic alteration in NSCLC is ALK fusions or rearrangements. Alectinib is preferred as first-line therapy in these patients. In the absence of significant toxicity, treatment is continued until there is evidence of progression. Ceritinib and brigatinib are other potent second-generation ALK inhibitors and are acceptable frontline alternatives to alectinib or in patients who have progressed on or are intolerant to crizotinib. Inhibitors for other genomic alterations in BRAF V600E, MET exon 14 splicing mutations, ERBB2 (HER2) mutations, KRAS G12C, and ROS1-, RET-, and NTRK-gene fusions are summarized in Figure 3. 

### 5.2. Selection of First-Line Systemic Therapy with PD-1/PD-L1 Inhibitor Monotherapy

First-generation ICIs include monoclonal antibodies (mAbs) targeting PD-1, such as nivolumab, pembrolizumab, cemiplimab, and toripalimab, as well as mAbs against PD-L1, such as atezolizumab, durvalumab, and avelumab. They also include mAbs against CTLA-4, such as ipilimumab and tremelimumab. Among these ICIs, pembrolizumab, either as monotherapy in patients with NSCLC whose tumors express high PD-L1 (≥50% total progression score, TPS) by IHC and lack EGFR or ALK genomic tumor aberrations, or in combination with pemetrexed and carboplatin in patients with non-squamous NSCLC whose tumors express no or low PD-L1 (i.e., 0–49% TPS), has been demonstrated to improve OS compared to standard-of-care chemotherapy as first-line systemic therapy [56,57,58,59,60,61]. According to the current NCCN guideline for NSCLC [33], contraindications for treatment with PD-1/PD-L1 inhibitors may include active or previously documented autoimmune disease and/or current use of immunosuppressive agents or the presence of an oncogene, which would predict a lack of benefit.

The KEYNOTE-024 study was the first to demonstrate an OS benefit with anti-PD-1 monotherapy compared to platinum-based chemotherapy as first-line treatment in patients with advanced NSCLC, thereby changing the treatment paradigm for this disease. In the updated analysis of KEYNOTE-024 [59], pembrolizumab continues to show an OS benefit as first-line therapy for advanced NSCLC with PD-L1 TPS of ≥50% compared to platinum-based chemotherapy (hazard ratio (HR): 0.63; 95% CI: 0.47 to 0.86; nominal *p* = 0.002). This improvement in OS, initially observed at the second interim analysis, persisted despite significant crossover to pembrolizumab in the chemotherapy arm. By intention-to-treat analysis, patients randomly assigned to pembrolizumab achieved a median OS of 30.0 months, compared to 14.2 months for patients in the chemotherapy arm [62]. At a 5-year follow-up, patients treated with pembrolizumab had improved 5-year OS rates compared with patients treated with platinum-based chemotherapy (31.9% vs. 16.3%). Of note, 39 patients received 35 cycles of pembrolizumab, 82.1% of whom were still alive at the data cutoff (approximately 5 years) [63].

The results from the KEYNOTE-042 (NCT02220894) study confirmed and expanded upon those from KEYNOTE-024 by demonstrating significantly improved OS with pembrolizumab compared to platinum-based chemotherapy. This benefit was observed not only in patients with PD-L1 TPS of ≥50% but also in those with PD-L1 TPS of ≥20% and ≥1% [64]. In the 5-year updated report [65], pembrolizumab monotherapy preserves superior OS (versus chemotherapy) regardless of PD-L1 TPS (for TPS ≥50%, HR [95% CI]: 0.68 [0.57 to 0.81]; for TPS ≥20%, 0.75 [0.64 to 0.87]; TPS ≥ 1%, 0.79 [0.70 to 0.89]). The estimated 5-year OS rates with pembrolizumab were 21.9%, 19.4%, and 16.6%, respectively. For the 102 (16%) patients with PD-L1 TPS ≥1% who completed 35 cycles (i.e., 2 years) of pembrolizumab monotherapy, the ORR was 84.3%. The ORR was 15.2% for 33 patients who received a second course of pembrolizumab at progression after completion of 35 cycles of pembrolizumab. Subsequently, atezolizumab and cemiplimab monotherapy also showed survival benefits compared to standard-of-care platinum-based chemotherapy. These results are summarized in Table 1. 

### 5.3. Selection of First-Line Therapy with PD-1/PD-L1 Inhibitor in Combination with Chemotherapy

Pembrolizumab plus platinum-based chemotherapy is a standard-of-care first-line therapy for patients with metastatic NSCLC [85]. First, pembrolizumab plus carboplatin and pemetrexed were investigated in patients with newly diagnosed, LA/mNSCLC without EGFR/ALK alterations, regardless of PD-L1 TPS in the randomized cohort G of the phase I/II KEYNOTE-021 (NCT02039674) study. This combination demonstrated improved ORR and PFS compared to carboplatin and pemetrexed alone [86], leading to its approval in the United States. The phase III KEYNOTE-189 (NCT02578680) study confirmed the clinical benefit of first-line pembrolizumab plus pemetrexed and platinum in patients with non-squamous metastatic NSCLC, regardless of PD-L1 tumor expression [87]. In the phase III KEYNOTE-407 (NCT02775435) study, pembrolizumab in combination with carboplatin and paclitaxel or nab-paclitaxel demonstrated improved OS and PFS compared with placebo plus carboplatin and paclitaxel or nab-paclitaxel in patients with squamous histology, regardless of PD-L1 expression [73]. Additionally, other anti-PD-1 or anti-PD-L1 antibodies have been evaluated in combination with chemotherapy or immunotherapy in patients with LA/mNSCLC as summarized in Table 1.

### 5.4. First-Line Systemic Therapy with ICIs and Bevacizumab

Bevacizumab, the first anti-VEGF monoclonal antibody, was FDA-approved for various cancer types including LA or metastatic non-squamous NSCLC in October 2006 [51]. Its influence on tumor vasculature and immune responses has been demonstrated, particularly in combination with an ICI, showing safe administration and revealing impacts on inflammation, lymphocyte trafficking, and immune regulation with ICIs [88]. The production of VEGF within the tumor microenvironment (TME) enhances the expression of PD-1 and other inhibitory checkpoints involved in CD8+ T-cell exhaustion. This effect can be reversed by anti-angiogenic agents targeting VEGF-A–VEGFR. The combination of anti-PD-1 antibodies with these agents has demonstrated a strong and synergistic antitumor effect, particularly in tumors that produce high levels of VEGF-A [89]. Currently, bevacizumab in combination with pembrolizumab, atezolizumab, or durvalumab (NCT02039674, NCT01633970, NCT02572687) is being tested in patients with NSCLC. Of note, the phase 3 IMpower150 study showed that atezolizumab plus bevacizumab plus chemotherapy significantly improved PFS (8.3 months vs. 6.8 months) and OS (19.2 months vs. 14.7 months) in patients with metastatic non-squamous NSCLC, regardless of PD-L1 expression and EGFR or ALK genetic alteration status [70]. Furthermore, atezolizumab with carboplatin and nab-paclitaxel improved median PFS but not OS compared with carboplatin plus nab-paclitaxel in patients with squamous NSCLC [75]. 

### 5.5. First-Line CTLA-4 and PD-1 Antibody Combination 

The combination of CTLA-4 and PD-1 inhibitors can enhance the activation of tumor-specific T cells and facilitate the conversion of a tumor microenvironment from ‘‘cold’’ to ‘‘hot’’ [90]. These dual ICIs have been shown to significantly increase response rates and improve survival in patients with LA/mNSCLC [91]. CheckMate-227 is a phase III trial investigating nivolumab and ipilimumab as first-line therapy in advanced NSCLC. In this trial, patients received nivolumab 3 mg/kg IV every 2 weeks plus ipilimumab 1 mg/kg IV every 6 weeks. This combination resulted in a significant improvement in the ORR, reaching 45.3%. Median PFS was 7.1 months in patients with a TMB ≥10 mutations per megabase (mut/Mb), compared to 2.6 months for those with a TMB <10 mut/Mb. The 10 mut/Mb cutoff also identified a subgroup of patients with better 6-month PFS. Subsequent analyses demonstrated that first-line treatment with nivolumab plus ipilimumab resulted in a longer duration of OS than chemotherapy in patients with NSCLC, regardless of PD-L1 expression levels [76]. 

Several additional studies have evaluated the effect of adding dual PD-1 and CTLA-4 inhibitors to first-line platinum-based chemotherapy in patients with LA/mNSCLC with no EGFR- or ALK-sensitive genomic alterations (Table 1). CheckMate-9LA was a phase III randomized study designed to assess whether a limited course (two cycles) of platinum-based first-line chemotherapy combined with dual PD-1 and CTLA-4 inhibitors could further improve clinical outcomes. At a median 13.2 months follow-up, the median OS was 15.6 months in the experimental group compared to 10.9 months in the control group. However, the incidence of grade 3–4 treatment-related toxicities, including immune-related adverse events (irAEs), was higher in the experimental arm compared to the controlled arm [79]. At the 4-year follow-up, patients treated with nivolumab plus ipilimumab and chemotherapy continued to show a durable OS benefit over those treated with chemotherapy alone (21% versus 16%), regardless of tumor PD-L1 expression and/or histology [80]. There remains an unmet need to understand the mechanisms of irAEs and to develop effective strategies to prevent and/or reduce the frequency and severity of these adverse events, especially with combinational ICIs. 

However, not all ICI monotherapy or chemotherapy combinations have yielded positive results. For instance, nivolumab monotherapy at 3 mg/kg every 2 weeks in the CheckMate 026 (NCT02041533) study did not lead to improvements in PFS or OS as a first-line systemic treatment for patients with advanced NSCLC and PD-L1 expression of 5% or greater [92]. It is important to note that cross-trial comparisons should be interpreted with caution due to differences in PD-L1 assays, including the use of different anti-PD-L1 antibodies and thresholds, between CheckMate 026 and the KEYNOTE-024 and KEYNOTE-042 studies. These variations may, in part, explain the divergent outcomes observed. Table 2 summarizes major negative results from the combination of immunotherapy to date.

## 6. Conclusions

In summary, the diagnostic algorithm and the selection of first-line systemic treatment for patients with nonresectable, LA/mNSCLC have evolved significantly over the past two decades. This paradigm shift has been translated into longer survival and better quality of life for these patients. We acknowledge several limitations in our review. First, due to the lack of head-to-head comparisons among different first-line chemoimmunotherapy regimens and various immune biomarker assays, we could not endorse any specific regimen. Further studies are needed to analyze the preference for these regimens by practicing clinicians. Second, the diagnosis and treatment of advanced NSCLC vary significantly across different countries and regions due to factors such as epidemiology, patient demographics, tumor biology, regional treatment availability, and economic conditions. Our review primarily focuses on data obtained from U.S. FDA approvals and professional practice guidelines for US practitioners and should be interpreted with caution when applied to other countries and regions. 

Currently, docetaxel alone or with ramucirumab is the standard of care for second-line systemic treatment for these patients. Many ongoing studies are evaluating optional subsequent treatments for patients with NSCLC beyond the first-line treatment. An unmet need of precision oncology is the selection of subsequent therapeutic regimens with maximal efficacy and minimal unwanted toxicity for individual cancer patients based on real-time assessment of their molecular and immune biomarkers at a given time during the disease course.

## Figures and Tables

**Figure 1 cancers-16-02350-f001:**
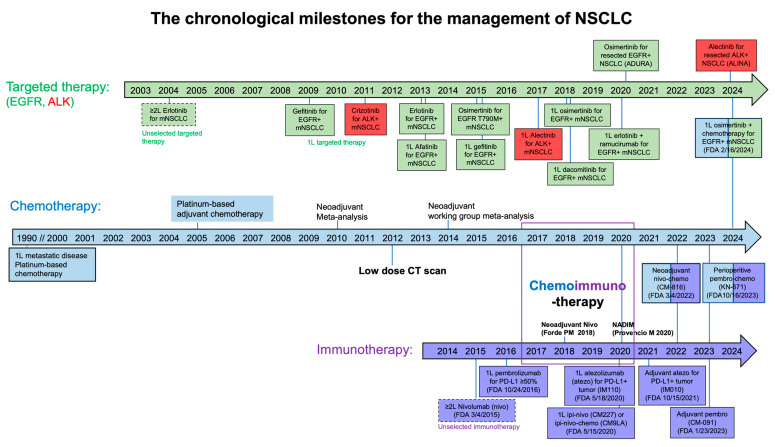
The chronological milestones for the management of NSCLC. Over the past few decades, many advances have contributed to the improved OS for patients with nonresectable, LA/mNSCLC, which includes first-line platinum-based combination chemotherapy, second-line single-agent chemotherapy or unselected molecularly targeted therapy, histology-directed chemotherapy, and tumor genotyping for molecular biomarkers, and first- and second-generation molecularly targeted therapies in the United States. Notably, patients with metastatic lung adenocarcinoma have benefited most from these therapeutic advances.

**Figure 2 cancers-16-02350-f002:**
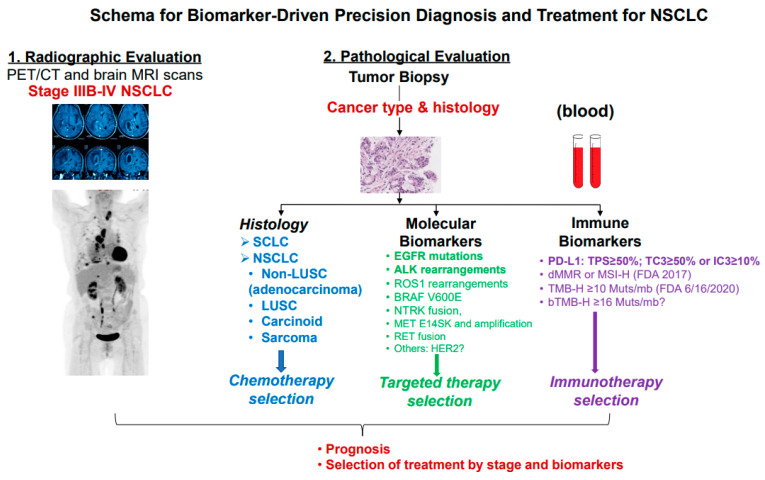
Schema for biomarker-driven precision diagnosis and treatment for NSCLC. This schema illustrates the clinical workflow, emphasizing the importance of multidisciplinary collaboration in implementing biomarker-driven precision diagnosis and treatment selection. It highlights the integration of histopathological, molecular, and immunological biomarkers at the time of NSCLC diagnosis to guide treatment selection. This holistic approach ensures that patients receive personalized therapy based on their individual biomarker profile, ultimately optimizing treatment outcomes.

**Figure 3 cancers-16-02350-f003:**
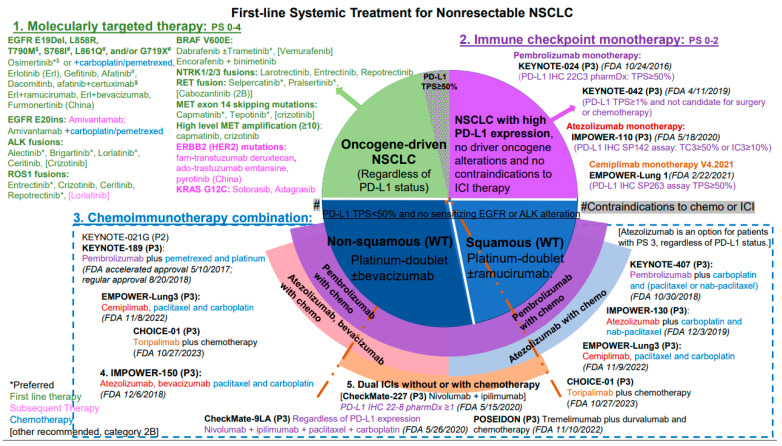
Schema for first-line systemic therapy for unresectable LA/mNSCLC. The landscape of first-line systemic treatment for patients with unresectable, LA/mNSCLC has evolved significantly over the past two decades. Compared to platinum-based chemotherapy doublets, molecularly targeted therapy and ICI therapy have prolonged survival with improved quality of life for the majority of patients with NSCLC. First, the detection of one of the 7 actionable oncogene alterations predicts a response rate of 60–80% to a matched TKI in approximately 25% of patients with NSCLC. Another 25% of patients with NSCLC benefit from a first-generation inhibitor against PD-1 or PD-L1 checkpoints. Thus, about 50% of patients with LA/mNSCLC do not need chemotherapy as first-line treatment. One exception to TKI targeted therapy is the NSCLC with osimertinib-resistant EGFR E20ins, for which the addition of dual EGFR and MET antibody amivantamab to carboplatin and pemetrexed was recently approved as the first-line treatment in March 2024. For those patients with oncogene-driven NSCLC, ICIs have not been shown to have any significant clinical activity. Several combination strategies of chemoimmunotherapy have been developed for the rest of ~50% of patients with NSCLC whose tumors do not have targetable genomic aberrations or high PD-L1 expression. Reference: 2024 NCCN V4, access date 28 April 2024.

**Table 1 cancers-16-02350-t001:** Summary of positive phase III trials of PD-1/PD-L1 inhibitors as first-line treatment for patients with advanced or metastatic NSCLC.

Strategy for Patient Population	Clinical Trial Identifier (Name)	Sample Size (*n*)	Biomarker	Assay (Vendor)	Experimental vs. Control Regimen	Median OS (mos)	Median PFS (mos)		Tumor Response (ORR, CR/PR)	Author (Year)
ICI monotherapy for stage IV or recurrent NSCLC with no sensitizing EGFR mutations or ALK translocations	NCT02142738 (KEYNOTE-024)	305	PD-L1 TPS ≥50%	22C3 (Dako)	Pembrolizumab 200 mg Q3W × 35 cycles vs. investigator’s choice of platinum-based CT × 4–6 cycles	30.0 vs. 14.2	10.3 vs. 6.0		44.8% vs. 27.8%	Reck (2016, 2019, 2021) [59,62,63]
						5-year OS: 26.3 vs. 13.4	7.7 vs. 5.5	5-year: 46.1% (4.5/41.6%) vs. 31.1% (0/31.1%)		
						5-year OS rate: 31.9% vs. 16.3%	PFS2: 24.1 vs. 8.5			
	NCT02220894 (KEYNOTE-042)	1274	PD-L1 TPS ≥ 50%	22C3 (Dako)	Pembrolizumab 200 mg Q3W × 35 cycles vs. investigator’s choice of platinum-based CT × 4–6 cycles	20.0 vs. 12.2	7.1 vs. 6.4		27.3% vs. 26.5%; 39.1% vs. 32.3%	Mok (2019 [64]; Castro (2022) [65]
			PD-L1 TPS ≥ 20%			17.7 vs. 13.0	6.2 vs. 6.6	33.2% vs. 29.1%		
			PD-L1 TPS ≥ 1%			16.7 vs. 12.1	5.4 vs. 6.5	27.3% vs. 26.7%		
			All cohorts			5-year OS rate: 21.9% vs. 19.4% vs. 16.6%		15.2% for retreatment		
	NCT02409342 (Impower-110)	572	High PD-L1 (TC3 ≥ 50% or IC3 ≥ 10%)	SP142 (VENTANA)	Atezolizumab 1200 mg Q3W vs. CT	20.2 vs. 13.1	8.1 vs. 5.0	68.3% vs. 35.7%	Herbst (2020) [66]	
			High + intermediate PD-L1 (≥5% of TC or IC)			18.2 vs. 14.9	7.2 vs. 5.5		30.7% vs. 32.1%; Any 29.2 vs. 31.8%	
	NCT03088540 (EMPOWER-Lung 1)	710	PD-L1 TPS ≥50%	SP263 (VENTANA)	Cemiplimab 350 mg Q3W vs. CT	NR vs. 14.2 (HR, 0.57)	8.2 vs. 5.7		39% (2/37) vs. 20% (1/19)	Sezer (2021) [67]
ICI-chemotherapy for non-squamous NSCLC with no EGFR or ALK genomic tumor aberrations		NCT02578680 (KEYNOTE-189)	616	PD-L1 TPS ≥1% vs. <1%	22C3 (Dako)	Pembrolizumab 200 mg Q3W vs. placebo × 35 + CT	NR vs. 11.3 (0.49); 1-year OS: 69.2% vs. 49.4%	8.8 vs. 4.9	47.6% vs. 18.9%	Gandhi (2018) [68]; Garassin o (2023) [69]
						22.0 vs. 10.6; 5-year OS rates: 19.4% vs. 11.3%	9.0 vs. 4.9; 5-year PFS rates: 7.5% vs. 0.6%			
	NCT02366143 (IMpower150)	1202	Teff wild-type (WT) population	T-effector gene signature; SP142 and SP263	Atezolizumab 1200 mg Q3W to control arm (ABCP) vs. BCP	WT: 19.2 vs. 14.7	WT: 8.3 vs. 6.8		63.5% vs. 48%	Socinski (2018, 2021) [70,71]
			Teff-high WT population			Teff-high: NA	Teff-high: 11.3 vs. 6.8			
	NCT02367781 (IMpower130)	724	PD-L1 TPS ≥1% or <1%	SP142 and SP263	Atezolizumab + CT vs. CT	18.6 vs. 13.9	7.0 vs. 5.5		49.2% vs. 31.9%	West (2019) [72]
ICI-chemotherapy for squamous NSCLC	NCT02775435 (KEYNOTE-407)	506	PD-L1 TPS ≥1% or <1%	22C3 (Dako)	Pembrolizumab + CT × 35 cycles vs. CT	7.8-month FU: 15.9 vs. 11.3	6.4 vs. 4.8		57.9% vs. 38.4%	Paz-Ares (2018, 2020) [73]; Novello (2023) [74]
						14-month FU: 17.1 vs. 11.6;	8.0 vs. 5.1			
						17.2 vs. 11.6; 5-year OS rates: 18.4% vs. 9.7%	8.0 vs. 5.1			
	NCT02367794 (IMpower 131)	1021	Tumor Gene Expression and PD-L1 TPS ≥1%, 10% or 50%	Gene profiling; SP142 and SP263	Atezolizumab + SoC Chemo vs. Soc Chemo × 4–6 cycles	14.2 vs. 13.5	6.3 vs. 5.6		49.7% vs. 41%	Jotte (2020) [75]
ICI-chemotherapy for NSCLC regardless of PD-L1 expression and histology subtypes	NCT02477826 (CheckMate 227)	1739	Co-Primary: PFS in TMB ≥10 muts/mb (*n* = 679)	FoundationOne CDx and 28-8 (Dako)	Nivolumab (nivo) 3 mg/kg IV Q2W and Ipilimumab (ipi) 1 mg/kg Q6W vs. CT	NA	7.2 vs. 5.5		45.3% vs. 26.9%	Hellmann (2018)[76]
			OS in PD-L1 TPS ≥1% or <1%			61-mo FU: 17.1 vs. 14.9 (PD-L1 ≥ 1%) and 17.4 vs. 12.2 (PD-L1 < 1%)	61-mo FU: 5.1 vs. 5.6 (PD-L1 ≥ 1%) and 5.1 vs. 4.7 (PD-L1 < 1%)		36% vs. 30% (PD-L1 ≥ 1%) and 27% vs. 23% (PD-L1 < 1%)	Hellmann (2019) [77]
			OS in PD-L1 TPS ≥1% or <1%			5-yr OS rates: 24% vs. 14% (PD-L1 ≥ 1%) and 19% vs. 7% (PD-L1 < 1%)			Brahmer (2022) [78]	
	NCT03215706 (CheckMate-9LA)	1150 enrolled, 719 randomized	Two co-primary endpoints: PD-L1 < 1% and PD-L1 ≥1%	28-8 pharmDx assay (Agilent Dako)	Nivo 360 mg Q3W + Ipi 1 mg/kg Q6W + CT × 2 cycles vs. CT	9.7-month FU: 14.1 vs. 10.3 After 2 more weeks; 15.6 vs. 10.9	6.8 vs. 5		37.7% vs. 25.1%	Paz-Arez (2021, 2023) [79]; Carbone (2024) [80]
						At the 4-year follow-up, OS 21% vs. 16%				
	NCT03409614 (EMPOWER-Lung 3)	466	Any level of PD-L1 expression	VENTANA PD-L1(SP263) assay	Cemiplimab 350 mg ×108 weeks + CT Q3W vs. CT	21.9 vs. 13.0	8.2 vs. 5.0		43.3% vs. 22.7%	Gogishvili (2022) [81]
	NCT NCT03164616 (POSEIDON)	1013	PD-L1 ≥50% of TCs or PD-L1 <50% of TCs	VENTANA PD-L1(SP263) assay	T 75 mg + D 1500 mg + CT × 4 and T×1	14.0	6.2		38.8%	Johnson (2022) [82]
					D+CT × 4, followed by D Q4W	13.3	5.5	41.5%		
					CT	11.7	4.8	24.4%		
	NCT03856411 (CHOICE-01)	465	TMB (*n* = 394), H-TMB (≥10 mutations/Mb)	PD-L1 (JS311 antibody) (MEDx, China)	Toripalimab 240 mg vs. Placebo + CT × 4-6, followed by toripalimab vs. placebo Q3W	Not reached vs. 17.1 (HR 0.69); 23.8 vs. 17.0	8.4 vs. 5.6; H-TMB: 13.1 vs. 5.5		65.7% vs. 46.2% H-TMB: 72.7% vs. 46.7%	Wang (2023) [83,84]

**Table 2 cancers-16-02350-t002:** Summary of negative first-line PD-1/PD-L1 inhibitors in patients with advanced or metastatic NSCLC.

Clinical Trial Identifier (Name)	Phase of Clinical Trial	Sample Size (No. Patients)	Biomarker	Assay (Marker)	Regimen	Control Regimen	Tumor Response (ORR)	Median PFS (mos) (HR)	Median OS (months)	Author (Year) Reference:
NCT02041533 (Checkmate 026)	Phase III	423	PD-L1 TPS ≥5% or <5%; TMB	28–8 (Dako)	Nivolumab 3 mg/kg IV Q2W	SoC platinum-based chemotherapy	26.6% vs. 33%	4.2 vs. 5.9 (1.5)	14.4 vs. 13.2	Carbone (2017) [92]
NCT02453282 (MYSTIC)	Phase III	1118	PD-L1 TPS ≥25% or <25%	SP263 (Ventana)	Durvalumab 10 mg/kg IV Q2W for up to 12 months and tremelimumab 1 mg/kg IV Q4W for up to 4 doses	SoC platinum-based chemotherapy	22.0%	NR (0.85)	NR	Rizvi (2018) [93]
NCT02395172 (JAVELIN Lung 200)	Phase III	396	PD-L1 PS ≥1% or <1%	73–10	Avelumab 10 mg/kg IV Q2W	Docetaxel 75 mg/m2 every 3 weeks.	15.0%	2.8 vs. 4.2 (0.9)	11.4 vs. 10.3	Barlesi (2018) [94]
NCT02879994	Phase II	25	PD-L1 TPS ≥50% or <50%; EGFR+	22C3 (Dako)	Pembrolizumab 200 mg IV Q3W	/	9.0%	/	/	Lsiberg (2018) [95]
NCT02087423 (ATLANTIC)	Phase II	111	PD-L1 TPS ≥25% or <25%; EGFR+ or ALK+	SP263 (Ventana)	Durvalumab 10 mg/kg IV Q2W (EGFR+/ALK+)	Durvalumab 10 mg/kg IV every 2 weeks (EGFR-/ALK-)	12.2% vs. 16.4%	1.9 vs. 3.3	13.3 vs. 10.9	Garassino (2018) [96]
CheckMate 370	Phase I	13	PD-L1 positive tumor cells; ALK+	28–8 (Dako)	Nivolumab 240 mg Q2W and crizotinib 250 mg twice daily	/	38.0%	/	/	Spigel (2018) [97]
NCT02143466 (TATTON)	Phase I	34	EGFR+	SP263 (Ventana)	Osimertinib 80 mg QD and durvalumab 3 mg/kg Q2W	/	PR: 53%	/	/	Anh (2016) [98]

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
