# Peer review of "Evolving Precision First-Line Systemic Treatment for Patients with Unresectable Non-Small Cell Lung Cancer"

_cancers, 2024, doi:10.3390/cancers16132350_

Round 1

Reviewer 1 Report

Comments and Suggestions for Authors

This manuscript presents a comprehensive and detailed review of the evolving landscape of first-line systemic treatments for patients with unresectable non-small cell lung cancer (NSCLC). Here are some specific suggestions for revisions:

1. In the title of the manuscript, to capitalize 'First-Line' and 'Non-Small Cell Lung Cancer' for consistency and clarity. 

2. The current summary is informative but slightly wordy and could be more concise.

3. Add a clear statement of the study's objectives at the end of the introduction. For example: "The objectives of this review are to summarize the current data on first-line systemic treatment for unresectable NSCLC and to propose a practical algorithm for implementing precision biomarker testing at diagnosis.

4.  Acknowledge any limitations or variability in the data that might affect the generalizability of the conclusions. For example: "While the proposed algorithm is based on robust data, it should be noted that variations in patient demographics and regional treatment availability may impact its applicability."

5. Identify gaps in the current data and suggest areas for further research to validate and refine the proposed algorithm.

6. The proposed algorithm is based on data and guidelines primarily from the United States, which may not be directly applicable to other regions with different healthcare infrastructures and patient demographics.

7. The conclusions are largely based on clinical trial data, which may not fully capture the complexities and variabilities seen in real-world clinical practice.

Author Response

Thank you very much for the enthusiastic review and insightful suggestions. Please see our point-by-point responses below.

1. In the title of the manuscript, to capitalize 'First-Line' and 'Non-Small Cell Lung Cancer' for consistency and clarity.

Response: done

2. The current summary is informative but slightly wordy and could be more concise.

Response: We have eliminated the sentences that have data summarized in Table 1.

3. Add a clear statement of the study's objectives at the end of the introduction. For example: "The objectives of this review are to summarize the current data on first-line systemic treatment for unresectable NSCLC and to propose a practical algorithm for implementing precision biomarker testing at diagnosis.

Response: We appreciated this suggestion and have added the objective in the abstract and at the end for first paragraph. “The objectives of this review are to summarize the current data and perspectives on first-line systemic treatment in patients with unresectable NSCLC and propose a practical algorithm for implementing precision biomarker testing at diagnosis.”

4. Acknowledge any limitations or variability in the data that might affect the generalizability of the conclusions. For example: "While the proposed algorithm is based on robust data, it should be noted that variations in patient demographics and regional treatment availability may impact its applicability."

Response: We agreed with this suggestion and have included the following sentences in the Summary: “We acknowledge several limitations in our review. First, due to the lack of head-to-head comparisons among different first-line chemoimmunotherapy regimens and various immune biomarker assays, we could not endorse any specific regimen. Further studies are needed to analyze the preference of these regimens by practicing clinicians. Second, the diagnosis and treatment of advanced NSCLC vary significantly across different countries and regions due to factors such as epidemiology, patient demographics, tumor biology, regional treatment availability, and economic conditions. Our review primarily focuses on data obtained from US FDA approvals and professional practice guidelines for US practitioners, and should be interpreted with caution when applied to other countries and regions.”

5. Identify gaps in the current data and suggest areas for further research to validate and refine the proposed algorithm.

Response: In addition to the above limitations, we address the following gap: “Currently, docetaxel alone or with ramucirumab is the standard of care for second-line systemic treatment for these patients. Many ongoing studies are evaluating optional subsequent treatments for NSCLC patients beyond the first-line treatment. An un-met need of precision oncology is the selection of subsequent therapeutic regimens with maximal efficacy and minimal unwanted toxicity for individual cancer patients based on real-time assessment of their molecular and immune biomarkers at a given time during the disease course.”

6. The proposed algorithm is based on data and guidelines primarily from the United States, which may not be directly applicable to other regions with different healthcare infrastructures and patient demographics.

Response: Please see the response in #4 above.  

7. The conclusions are largely based on clinical trial data, which may not fully capture the complexities and variabilities seen in real-world clinical practice

Response: We appreciated this suggestion and added this sentence in the last paragraph. “We acknowledge several limitations in our review. First, due to the lack of head-to-head comparisons among different first-line chemoimmunotherapy regimens and various immune biomarker assays, we could not endorse any specific regimen. Further studies are needed to analyze the preference of these regimens by practicing clinicians.”

Reviewer 2 Report

Comments and Suggestions for Authors

1. The reason for the increase in lung cancer is thought to be the aging of people who used to smoke. Currently, smoking cessation is mainstream, but this is only for the current younger generation, and lung cancer is thought to be on the decline in the future. It would be better to include such a statement.

2. It is unclear what the final goal of this review is, so it should be made clear.

3. The combination of therapeutic drugs is known, and many doctors probably already understand the contents of this paper.

4. Since it is not clear what results have been seen in actual human trials, it is necessary to state how many people improved, what the 5-year survival rate was, and what the complete recovery rate was, rather than just explaining the medicine.

Although the content is very interesting, this content is a review and is not sufficient for publication in a cancer journal.

Author Response

Thank you very much for the review and insightful suggestions. Please see our point-by-point responses below.

1. The reason for the increase in lung cancer is thought to be the aging of people who used to smoke. Currently, smoking cessation is mainstream, but this is only for the current younger generation, and lung cancer is thought to be on the decline in the future. It would be better to include such a statement.

Response: We agreed with the Reviewer, and have included the following sentences in the first paragraph: “In the United States, although lung cancer incidence has steadily declined since 2006 by 2.5% annually in men and by 1% annually in women, lung cancer remains the leading cause of cancer death in both males and women [2]. The survival rate of lung cancer differs by histology subtype, stage at diagnosis, access to care, and treatment, which have significantly geographic variation. …. Despite of the increasing impact of smoking cessation, environmental control, and early detection, currently over 40% of NSCLC patients are diagnosed with distant metastasis with palliative systemic therapy as the main treatment option [2].”

2. It is unclear what the final goal of this review is, so it should be made clear.

Response: We appreciated this suggestion and have added the objective in the abstract and at the end for first paragraph. “The objectives of this review are to summarize the current data on first-line systemic treatment for unresectable NSCLC and to propose a practical algorithm for implementing precision biomarker testing at diagnosis.”

3. The combination of therapeutic drugs is known, and many doctors probably already understand the contents of this paper.

Response: Although the trials in this review have been reported before, we have included recent updates on these studies in Table 1, and summarized the data in the Figures which will help interpreting the results of evolving first line systemic therapies.

4. Since it is not clear what results have been seen in actual human trials, it is necessary to state how many people improved, what the 5-year survival rate was, and what the complete recovery rate was, rather than just explaining the medicine. Although the content is very interesting, this content is a review and is not sufficient for publication in a cancer journal.

Response: This review was commissioned by the Journal. The key trial information, which includes patient number, treatment, and key results (including available 5-year survival rates) are summarized in Table 1.

Round 2

Reviewer 2 Report

Comments and Suggestions for Authors

There should be no problem with this manuscript.